# Fructose and the Liver

**DOI:** 10.3390/ijms22136969

**Published:** 2021-06-28

**Authors:** Pablo Muriel, Pedro López-Sánchez, Erika Ramos-Tovar

**Affiliations:** 1Laboratory of Experimental Hepatology, Department of Pharmacology, Cinvestav-IPN, Apartado Postal 14-740, Mexico City 07300, Mexico; pmuriel@cinvestav.mx; 2Postgraduate Studies and Research Section, School of Higher Education in Medicine-IPN, Plan de San Luis y Díaz Mirón s/n, Casco de Santo Tomás, Mexico City 11340, Mexico; pelosa651018@yahoo.com

**Keywords:** liver, fructose, uric acid, NLRP3, oxidative stress, inflammation

## Abstract

Chronic diseases represent a major challenge in world health. Metabolic syndrome is a constellation of disturbances affecting several organs, and it has been proposed to be a liver-centered condition. Fructose overconsumption may result in insulin resistance, oxidative stress, inflammation, elevated uric acid levels, increased blood pressure, and increased triglyceride concentrations in both the blood and liver. Non-alcoholic fatty liver disease (NAFLD) is a term widely used to describe excessive fatty infiltration in the liver in the absence of alcohol, autoimmune disorders, or viral hepatitis; it is attributed to obesity, high sugar and fat consumption, and sedentarism. If untreated, NAFLD can progress to nonalcoholic steatohepatitis (NASH), characterized by inflammation and mild fibrosis in addition to fat infiltration and, eventually, advanced scar tissue deposition, cirrhosis, and finally liver cancer, which constitutes the culmination of the disease. Notably, fructose is recognized as a major mediator of NAFLD, as a significant correlation between fructose intake and the degree of inflammation and fibrosis has been found in preclinical and clinical studies. Moreover, fructose is a risk factor for liver cancer development. Interestingly, fructose induces a number of proinflammatory, fibrogenic, and oncogenic signaling pathways that explain its deleterious effects in the body, especially in the liver.

## 1. Introduction

Chronic diseases represent a major challenge in world health. Metabolic syndrome is a constellation of disturbances that includes dyslipidemia, type II diabetes, insulin resistance, visceral obesity, microalbuminuria, and hypertension [1,2]. The prevalence of metabolic syndrome is difficult to establish because there is no consensus on its definition [1], but estimations are 27.93% in North America, 27.65% in South America, 21.27% in Asia, 16.04% in Africa, and 10.47% in Europe [3], affecting a quarter of the world’s population [4]. The most important risk factors for developing metabolic syndrome are related to obesity, a complex disease associated with an imbalance between physical activity and calorie intake, and excessive consumption of fats and simple carbohydrates; the obesogenic environment also plays an important role [5]. Approximately one-third of adults, children, or adolescents worldwide are obese or overweight [1,2,6].

Metabolic syndrome affects several organs, and it has been proposed to be a liver-centered condition [7]. Non-alcoholic fatty liver disease (NAFLD) is a term widely used to describe excessive fat infiltration in the liver in the absence of alcohol, autoimmune disorders, and viral hepatitis [6]. NAFLD now constitutes the main cause of hepatic disorders. It is usually asymptomatic, bidirectionally linked with metabolic syndrome, and difficult to diagnose, affecting about a third of the global population, and it is the prevailing cause of hepatocellular carcinoma (HCC) development [8,9]. Thirty percent of NAFLD patients develop necroinflammation and fibrosis, indicating the presence of nonalcoholic steatohepatitis (NASH), which in turn may predispose patients to HCC [10,11,12,13]. Moreover, NASH is a risk factor for liver cancer development [2,14,15,16,17,18]. HCC is the dominant form of primary liver cancer, which represents 75–90% of the total liver cancer burden [19,20]. In the early stages, HCC lacks symptoms and exhibits a rapid growth of malignant cells, resulting in the late diagnosis of the disease, and, therefore, at least one-quarter of HCC cases remain idiopathic and can be attributed to NAFLD [21,22,23]. Unfortunately, no effective therapy for NAFLD is currently available, and there is no scientific evidence to recommend specific diets for this group of patients. This paper provides a review of fructose deleterious effects on the liver and describes the molecular mechanisms involved.

Based on animal model experiments and clinical studies, fructose is recognized as a major mediator of NAFLD [24,25,26]. A significant correlation between fructose intake and the degree of fibrosis has been found [24,27]. Fructose is present in fruits and table sugar (sucrose, which is 50% glucose and 50% fructose), and has a sweeter taste and lower glycemic index than glucose. Its consumption has recently increased in many parts of the world because of the growing use of high-fructose corn syrup in beverages and processed food [21]. Studies on ancestral diets have shown that the average intake of fructose per capita was around 2 kg per year, while the current global average consumption of fructose per capita is 25 kg per year [28]. High-fructose corn syrup is made from corn using caustic soda, hydrochloric acid, and enzymes and is classified according to the percentage of fructose content (90, 55, and 42%). This powerful and cheap sweetener provides a long shelf life and maintains long-lasting hydration in industrial bakeries [29]. In developing countries, such as Mexico, its importation from the United States has significantly increased recently, indicating a larger demand for added sugars in these emerging markets. Sugar-sweetened beverages provide 60% of the daily sugar intake in the United States. Mexico has the largest number of sugar-sweetened beverage consumers worldwide, where such beverages provide 69% of the total added sugar in the daily diet [30,31]. Additionally, NAFLD patients consume twice as many calories from beverages sweetened with high-fructose corn syrup as healthy patients [32]. The World Health Organization recommends reducing the intake of free sugars to less than 10% of the total daily energy intake because of its association with metabolic diseases and cancer [32,33,34]. Furthermore, elevated consumption of fructose represents a great metabolic risk for not only obese but also lean individuals who have a high consumption of fructose-sweetened beverages [7,24,35].

## 2. Deleterious Metabolic Effects of Fructose

### 2.1. The Initial Physiological Impact of High Fructose Consumption

Fructose possesses an open-chain chemical conformation and is therefore much more reactive than glucose [36]. Experimental studies have shown that a high fructose intake promotes oxidative stress, inflammation, higher serum uric acid levels, hypertriglyceridemia, higher systolic blood pressure, and insulin resistance [37,38] (Figure 1). In humans, the physiological impact depends on the formulation in which the fructose is consumed; consumption via solids and liquids differently affects microbiota composition, gut integrity, and liver toxicity [39,40].

Sensory stimulation is the adaptive response to food intake through rapid physiological processes, and one of the most studied is the cephalic-phase insulin response. Oral fructose stimulates autonomic and endocrine responses, which downregulate the cephalic phase of the insulin pathway in taste cells, reducing pancreatic insulin production [41]. Additionally, eating fructose, in contrast to glucose consumption, leads to increased hunger and desire to eat because fructose decreases leptin and glucagon-like peptide 1 and increases ghrelin levels in the serum [42]. Ghrelin activates the neuronal activity of neuropeptide Y, increasing food intake, and glucagon-like peptide 1 inhibition causes a decrease in insulin secretion [43]. Increased dietary fructose intake significantly accelerated the half-emptying time in the stomach compared to a similar intake of glucose [44]. Fructose, in the mouth and gut, may impact eating behavior by sweet-tasting mechanisms [45]. Sweet foods have powerful reinforcing effects mediated, in part, by dopamine receptors and, on vulnerable individuals, may overwhelm the homeostatic control mechanisms of the brain, possibly inducing behavioral alterations observed in addiction, such as anxiety or craving [46,47,48]. Regarding the hedonic value of fructose and the sum of all these events that affect appetite control, more studies are required to understand the role of fructose in the reward system.

### 2.2. Fructose in the Intestine

The intestinal epithelium is the cell layer closest to the intestinal lumen and is composed by 70–80% of enterocytes, Paneth cells, goblet cells, and intestinal stem cells. Studies attribute the metabolic effects of fructose to enterocytes, cells specialized in absorption [49].

#### 2.2.1. Intestinal Absorption of Fructose

The human small intestine expresses all the fructose-metabolizing enzymes; glucose transporter protein member 5 (Glut5) is the main protein responsible for the absorption of fructose into the cytosol [50,51]. Glut5, which mediates the active transport of fructose in mice, is mainly found in the small intestine [52]. Glut2 (SLC2A2) is a non-specific glucose transporter expressed in the intestinal basolateral membrane, which transports fructose by a facilitated passive mechanism from the gut into the hepatic portal circulation [53]. In humans, fructose is converted to glucose when the intake is moderate (≤1 g/kg of body weight), while high fructose consumption leads to the strong induction of Glut5 but not Glut2, thus increasing the fructose concentration and catabolism in the cytosol of intestinal epithelial cells [36,54,55,56]. The deletion of Glut5 in mice has been shown to reduce fructose absorption [57]. Glut5 is a transceptor, a transporter that binds to its substrate and activates intracellular signaling that triggers multiple responses [58]. Thioredoxin-interacting protein (TXNIP) is a multifunctional intracellular protein that coordinates signaling pathways during oxidative stress and inflammation [59]. TXNIP is also a regulator of carbohydrate metabolism [60]. Glut5 binds to TXNIP, which leads to increased Glut5 gene expression and protein synthesis, facilitating its migration to the apical membrane, thus improving fructose absorption [59]. In the cytoplasm of intestinal cells, the ketohexokinase (KHK) enzyme, also called fructokinase, which has a high affinity for fructose, phosphorylates fructose to fructose-1-phosphate, a toxic metabolite [61]. Excess phosphorylated fructose is conducted to the hexosamine pathway, which increases O-glucosamine-N-acetyl transferase activity and consequently upregulates the expression of transforming growth factor-beta (TGF-β) [62]. There are two isoforms of KHK, KHK-C and KHK-A; the latter has a ten-times-lower affinity for fructose than KHK-C and therefore consumes ATP more slowly [7,63]. KHK-A may decrease fructose metabolism in the liver and, thus, may inhibit the development of metabolic syndrome [36]. By contrast, KHK-C overexpression promotes intestinal fructose clearance and increases fructose-induced lipogenesis in the liver [61]. However, when the capacity for intestinal fructose clearance is exceeded, the increased activity of KHK-C exhausts adenosine triphosphate (ATP) and induces adenosine monophosphate deaminase activation, which results in marked ATP depletion, leading to the accumulation of adenosine monophosphate and uric acid production [64]. Preclinical evidence using human livers, KHK inhibition to improve steatosis, inflammation and fibrosis in NAFLD [65].

#### 2.2.2. Intestinal Production of Uric Acid by Fructose

Uric acid, a weak organic acid end-product of purine catabolism in humans, is an antioxidant molecule that plays an essential role in the cardiac, vascular, and central nervous systems because it can neutralize pro-oxidant free radicals, such as hydroxyl radicals, hydrogen peroxide, and peroxynitrite [64]. Uric acid is produced in the liver and gut and excreted through the urine and feces [64]. According to Yun et al., the duodenum plays an important role in the synthesis and elimination of uric acid; one-third of the total uric acid is excreted through the gut [66]. Thirty percent of uric acid is excreted via ATP-binding cassette subfamily 2 (breast cancer resistance protein) on the luminal surface of the intestine, but an imbalance in its production or excretion can increase uric acid levels, favoring nicotinamide adenine dinucleotide (NADPH) oxidase (NOX) activation in the liver, acting as a damage-associated molecular pattern (DAMP) [67,68].

#### 2.2.3. Fructose Induces Lipogenesis and Oxidative Stress in the Intestine

Furthermore, a high fructose intake in an experimental model can activate carbohydrate-responsive element-binding protein (ChREBP) and sterol-responsive element-binding protein (SREBP), which induce fructolytic and lipogenic enzymes, respectively [69]. ChREBP is a transcription factor activated by a high-fructose diet, improving the KHK and Glut5 capacity for fructose absorption [70]. SREBP is a family of transcription factors consisting of three isoforms that regulate the homeostasis of lipids. In enterocytes, apolipoprotein induces the transcription of SREBP1c, which improves the stability of ApoB-48, the structural protein for chylomicrons, enhances microsomal triglyceride transfer protein, and augments lipogenesis [69]. This uncontrolled lipid metabolism and lower clearance of chylomicrons in the intestinal cells, together with uric acid overproduction, is responsible for increased cardiometabolic risk and leads to the development of NASH [70,71,72].

NASH models showed that cytochrome P450 2E1 activity is linked to increased intestinal inflammation during fructose consumption [73]. Cytochrome P450 2E1 plays a critical role in the metabolism of fatty acids. Furthermore, NASH patients have increased cytochrome P450-2E1 and inducible nitric oxide synthase, which cause the nitration of intestinal tight and adherent junction proteins [74]. The disruption of tight junction proteins and elevated apoptosis of enterocytes, evidenced by the upregulation of caspase 3 and p-JNK after fructose exposure, contributes to endoplasmic reticulum stress, the accumulation of unfolded or misfolded proteins, and the dysfunction of the epithelial barrier, which result in increased gut permeability, allowing lipopolysaccharides (LPS) to translocate from the gut lumen to the portal tract, triggering an inflammatory response in the liver [74]. Ca^2+^ absorption is one of the most important intestinal functions, and glutathione (GSH) is essential for this process [75]. γ-L-glutamyl-L-cysteinylglycine, or GSH, is the main intracellular cofactor protecting against oxidative stress in the gut, and its biosynthesis occurs in the cytosol through ATP-dependent reactions [76]. The antioxidant activity of GSH is catalyzed by GSH peroxidase (GPx), which reduces hydrogen peroxide and lipid peroxides as GSH is oxidized to GSSG [77]. In animal models that use fructose-rich diets, the intestinal absorption of Ca^2+^ is decreased, and Ca^2+^ receptors are depleted, which leads to decreased antioxidant defenses (GPx, catalase, superoxide dismutase, etc., are exhausted), and endoplasmic reticulum stress occurs [75]. Similarly, increased fructose phosphorylation triggers ATP depletion, as mentioned earlier, inhibiting GSH restoration.

#### 2.2.4. Fructose and the Microbiota

The composition and function of the microbiota are regulated by multiple factors, such as diet and physical activity. Recent reports show that fructose consumption alters the gut microbiota and their bacterial metabolites, in a manner that promotes the development and progression of NASH [78]. Excessive fructose consumption decreases the expression of intestinal tight junction proteins, such as zonula occludens 1, junctional adhesion molecule A, occludin, claudin, β-catenin, and E-cadherin [74,79]. This environment generates dysbiosis by increasing *Bacteroides, Proteobacteria, Enterobacteria, Escherichia, Blautia producta*, and *Bacteroides fragilis* while decreasing *Actinobacteria, Akkermansia, Verrucomicrobia, Coprococcus eutactus,* and *Lactobacillus*, increasing the loss and blebbing of the laminar propria, which triggers inflammation in the small intestine, and, due to the increase in gut permeability, toxic bacterial metabolites may reach the liver, contributing to inflammation in NASH [29,36,74,80,81]. Similarly, diets enriched with fructose alter the composition of the short-chain fatty acids in the gut, inducing a high microbial production of butyrate, acetate or propionate by the intestinal microbiota, therefore increasing the production of acetyl-CoA from acetate, which contributes to lipogenesis [82]. Ethanol is also an important fructose metabolite that has been associated with NAFLD. Patients suffering from NAFLD who abuse alcohol exhibit more severe liver injury than those with any of these factors individually [83]. It is noteworthy that *Escherichia, Bacteroides*, and *Clostridium* bacteria can produce ethanol. In patients with NAFLD, the activity of alcohol-metabolizing enzymes, such as alcohol dehydrogenase, and the microbiota are dysregulated [84]. As a consequence, increased blood ethanol concentrations and/or ethanol metabolites can alter the host’s metabolism, generate reactive oxygen species, and active inflammatory pathways, suggesting that microbiota that produce alcohol can have important effects on the evolution of NAFLD [85,86,87]. Moreover, gut dysbiosis triggered by excessive fructose intake leads to intestinal bacterial overgrowth, a strong decrease in microbial diversity, and increased translocation of bacterial products and cytotoxins, stimulating inflammatory pathways in experimental and human NAFLD [88,89] (Figure 2). These results indicate that high fructose in the intestine plays a major role in NAFLD development. The dysregulated microbiota, disruption of intestinal tight junction proteins, elevated uric acid production, and toxic bacterial metabolites accelerate NASH progression. The deleterious effects of fructose in the intestine could be ameliorated by the development of selective inhibitors of KHK-C, the limiting enzyme in fructose metabolism.

### 2.3. Fructose in the Liver

In humans, 70% of fructose is metabolized by the liver [90]. A diet rich in fructose induces the hepatic de novo synthesis of fatty acids and triglyceride accumulation [7,38,90]. Therefore, fructose has been postulated as a key factor for the development of NASH. Once fructose exceeds the intestinal clearance capacity, it is driven to the portal vein, where a fructosemic state strongly and quickly induces mechanisms involved in its overflow to the liver, which is the principal organ for fructose metabolism [7,38]. However, the mechanisms of the hepatic cell types (hepatocytes, hepatic stellate cells (HSCs), and Kupffer cells) that are involved in the metabolism of fructose consumed in large quantities are poorly understood [69]. In the liver, fructose is catabolized faster and is more lipogenic than glucose. In particular, chronic high fructose consumption induces the aldolase B enzyme, which breaks down fructose to dihydroxyacetone phosphate and D-glyceraldehyde. Then, triokinase stimulates the phosphorylation of D-glyceraldehyde to produce pyruvate and acetyl-CoA, promoting lipid dysregulation [36,54,91] (Figure 3).

#### 2.3.1. Ketohexokinase and Fructose

The liver plays the most important role in carbohydrate metabolism. The principal isoform of KHK in the liver is KHK-C, which phosphorylates fructose rapidly and without any negative feedback control. Similar to in mice, KHK expression is elevated in obese patients with advanced liver disease compared to in obese subjects without fatty liver [81]. In humans, KHK inhibition has been demonstrated to improve steatosis, ballooning degeneration, inflammation, and fibrosis in the liver [92]. In KHK-knockout mice, ATP citrate lyase (ACLY), acetyl-CoA carboxylase (ACC)-1, and fatty acid synthase (FASN) are decreased by fructose administration [81]. ACLY is an enzyme that links carbohydrate to lipid metabolism by converting citrate to acetyl-CoA for fatty acid and cholesterol biosynthesis. ACLY inhibition protects against hepatic steatosis, dyslipidemia, and associated complications such as atherosclerosis [93]. ACC-1 coordinates the synthesis of fatty acids in the liver and generates a pool of malonyl-CoA used by FASN to generate palmitate [94]. ACC-1 inhibition reduces lipotoxicity in hepatocytes and prevents HSC activation, which significantly reduces fibrosis in NASH [94] (Figure 3).

#### 2.3.2. Toll-Like Receptor-4 and Fructose

Kupffer cells play a central role in liver damage induced by fructose. The elevated endotoxemia and oxidative stress produced by fructose intake promote hepatic Toll-like receptor (TLR)-4 activation. The TLR-4/MyD88 signaling pathway in liver parenchymal cells plays a pivotal role during NASH development [85,87]. As previously mentioned, fructose causes gut-barrier deterioration through the disruption of tight-junction proteins. Endotoxins produced by Gram-negative bacteria alter intestinal permeability and cause bacterial translocation. Mice chronically fed fructose were found to have increased levels of endotoxins in portal blood and unregulated inflammatory mediators in Kupffer cells [95]. LPS and other bacterial toxins cross the gut barrier and bind to TLR-4 on the macrophages or Kupffer cells’ plasma membranes, which activates the proinflammatory signaling pathway, with a consequent increase in the expression of proinflammatory cytokines including tumor necrosis factor-alpha (TNF-α), interleukin (IL)-6, and IL-1beta (β) [78,96]. Specifically, the binding of LPS to TLR-4 on macrophages activates the nuclear factor kappa B (NF-κB) signaling pathway via the adaptor protein MyD88 and induces TNF-α expression and secretion [97]. TNF-α also stimulates fructose-driven steatosis in the human liver and induces the expression of the SREBP1-regulated enzymes ACC-1, FASN, and SREBP1c at the mRNA level [97]. The NF-κB/MyD88 pathway drives inflammasome activation, which is a cytosolic regulator of inflammation that, through the caspase-2 pathway, activates SREBP1c to induce ACC-1 and FASN, contributing to the exacerbation of hepatic steatosis and inflammation in NAFLD [97]. The deleterious mechanism induced by the binding of cytotoxic bacterial metabolites to TLR-4 is shown in Figure 4.

TLR-4 promotes NF-κB signaling, and this pathway upregulates the transcription of the NOD-like receptor family pyrin domain containing 3 (NLRP3) inflammasome and proinflammatory cytokines such as IL-1β and TNF-α [96,98]. Studies performed in mice models have shown that fructose triggers the infiltration/activation of macrophages/Kupffer cells, causing increased levels of ROS, and induces the necrosis of hepatocytes through TNF-α and IL-6 upregulation (90). The factors underlying the progression from NAFLD to NASH are multifactorial, but NLRP3 inflammasome activation is critically important. Cytokine release causes hepatocyte death along with activation of HSCs and Kupffer cells. The NLRP3 inflammasome, upregulated by fructose overfeeding, is a sensor of danger signals, DAMPs, uric acid crystals, or derivatives that act like DAMP molecules and induce inflammation [99,100,101]. The NLRP3 inflammasome recruits apoptosis-associated speck-like protein and pro-caspase 1, leading to the maturation and secretion of IL-1β and IL-18 [102,103]. Caspase 1 is necessary for the activation of the NLRP3 inflammasome, as an executioner molecule; then, IL-1β is matured, triggering HSC activation, and thus, fibrogenesis ensues [104]. Indeed, the levels of IL-1β correlate with the mRNA of collagen 1, a key profibrogenic gene [105,106]. The activation of the NLRP3 inflammasome is a synchronized interaction between hepatocytes and Kupffer cells that results in dyslipidemia and lipid accumulation in hepatocytes [107]. High fructose administration to rodents increases TXNIP levels and malondialdehyde and decreases superoxide dismutase, triggering oxidative stress, which is sensed by TXNIP, therefore inducing NLRP3 inflammasome activation [103]. The fructose–ROS–TXNIP–NLRP3 inflammasome axis is crucial in the pathogenesis of uric-acid-induced inflammatory responses [108] (Figure 5).

Wree et al. found that NLRP3 inflammasome activation results in severe liver inflammation and fibrosis via the pyroptotic signaling pathway in hepatocytes [109]. Pyroptosis is a unique form of programed cell death where a plasma membrane pore formed by gasdermin D allows the release of the cellular content, leading to the upregulation of proinflammatory cytokines and profibrogenic factors such as IL-1β, connective tissue growth factor, and TGF-β, triggering the activation of HSCs, leading to the increased production and secretion of scar tissue proteins [109]; as a result, inflammation is exacerbated and liver fibrosis ensues [103,110]. Inflammasome activation by fructose could also be the result of increased Glut5 activity, which induces TXNIP to form the activated complex of ASC with NLRP3, consequently inducing dyslipidemia, hepatic inflammation, and lipid accumulation [111]. In addition, there is evidence indicating that TXNIP is upregulated in the liver by the master nutritional regulator ChREBP [112].

#### 2.3.3. Nuclear Factor E2-Related Factor 2 and Fructose

Increasing evidence indicates that nuclear factor E2-related factor 2 (Nrf2) plays a complex, multicellular role within the processes of liver inflammation and fibrosis through the induction of its target genes [113,114]. Nrf2 is considered to act as the first line of defense against cellular damage due to oxidative stress [115]. It upregulates the expression of protective and antioxidant genes, upregulating the GSH biosynthesis and thioredoxin systems, to maintain cellular redox homeostasis in response to oxidative stress and other insults; therefore, its inactivation can exacerbate oxidant, inflammatory, and profibrotic processes [113,116,117]. Interestingly, oxidative stress, inflammation, and fibrosis are linked by several molecular signaling pathways that have been recently reviewed elsewhere [110]. The cytoplasmic protein repressor Kelch-like ECH-associated protein-1 (Keap1) regulates Nrf2′s function [110]. Keap1 acts as a sensor for oxidative stress, and under stress conditions, the sequestration complex dissociates, allowing Nrf2 to translocate to the nucleus, where it binds to the antioxidant response element and induces the expression of a battery of antioxidant genes [110]. In the liver, the activation of Nrf2 attenuates injuries of diverse etiologies, including chronic diseases such as NAFLD, by inducing heme oxygenase-1 (HO-1) expression and improving GSH efficacy [116,117]. Nrf2 activation prevents metabolic dysregulation and insulin resistance in mice through the repression of hepatic enzymes such as FASN and ACC and protects against hypertriglyceridemia and fatty liver disease; this protection is abolished when Nrf2 is deleted [118]. Acute fructose intake upregulates the expression of Nrf2 pathways, but excessive consumption through high-fructose diets increases reactive species and oxidative damage and downregulates Nrf2 and GSH [119,120]. MiRNAs are non-coding RNAs that regulate genes, silencing or promoting their expression through modulating mRNA transcription. MicroRNA (miRNA)-200a is reported to target Keap1, thereby activating Nrf2, and high fructose decreases miRNA-200a, which inhibits the Nrf2 antioxidant response [121]. The inhibition of KHK in the presence of fructose is accompanied by an increase in Nrf2 and the cytoprotective expression of HO-1, NAD(P)H dehydrogenase (quinone) 1 (NQO-1), and thioredoxin reductase 1 [92,117]. Mice deficient in Glut8 (SLC2A8), a member of the facilitated hexose transporter superfamily, have impaired hepatic first-pass fructose metabolism [122]. Transcriptomic analysis reveals that the excessive consumption of fructose induces mechanisms that increase oxidative stress, such as aryl hydrocarbon receptor downregulation. The aryl hydrocarbon receptor modulates the expression of various biotransformation enzymes classified as phase I and II enzymes; this receptor also has crosstalk with NF-κB [123]. Therefore, fructose intake, which causes the downregulation of xenobiotic-metabolizing enzymes and Nrf2 transcription, also leads to the upregulation of NF-κB [124,125,126].

#### 2.3.4. Carbohydrate Responsive Element-Binding Protein and Fructose

ChREBP is an essential transcription factor involved in hepatic stress that upregulates the ACLY, ACC-1, and FASN enzymes involved in hepatic de novo lipogenesis and, therefore, is a central factor in NAFLD [127,128]. However, the liver-specific deletion of ACLY fails to suppress fructose-induced lipogenesis [82]. By contrast, ACC-1 inhibition was associated with a decrease in hepatic de novo lipogenesis and insulin resistance and increased fatty acid β-oxidation [94]. Moreover, the inhibition of ACC-1 reduced the activation of TGF-β and fibrogenesis because HSC activation requires this factor and de novo lipogenesis [94]. The liver-specific ablation of ChREBP in rodents fed an elevated-fructose diet causes severe transaminitis and hepatomegaly with glycogen accumulation [129]. In addition, ChREBP induces the expression of fibroblast growth factor 21 (FGF21), which ameliorates dyslipidemia in humans [129]. FGF21 activates lipolysis and increases fatty acid oxidation in the liver through the activation of peroxisome proliferator-activated receptor alpha (PPAR-α). At the molecular level, these changes were associated with increases in the liver X receptor, which increases SREBP and decreases PPAR-α activation [130]. In humans, the expression of PPAR-α negatively correlates with the presence of NAFLD and the severity of steatosis [131]. PPAR-α, which is mainly activated during the fasted state and regulates the metabolism of lipids and inflammation, is primarily found in hepatocytes, and fatty acids resulting from the metabolism of fructose are oxidized to produce acetyl-CoA by peroxisomes and mitochondria through PPAR-α [76]. PPAR-α also stimulates the mitochondrial β-oxidation pathway and induces inhibitor kappa B (IκB)α in hepatocytes, which prevents the translocation of nuclear transcription factor kappa B (NF-κB) to the nucleus, a well-known proinflammatory signaler [78,96]. IκBα upregulates lipid metabolism and reduces inflammation, which improves NASH pathology [132]. By contrast, in FGF21-knockout mice, the activation of HSCs and fibrogenesis were increased, evidenced by increased levels of TGF-β, matrix metalloproteinases, and tissue inhibitors of metalloproteinases [129].

The respiratory chain of the mitochondria produces ROS, but ROS are decreased by antioxidant enzymes to prevent the deleterious effects of free radicals on important biological molecules. Long-term elevated fructose intake produces oxidative alterations in liver cells, particularly in the lipid components of mitochondria, and diminished superoxide dismutase and catalase activities, which are important enzymes for counteracting mitochondrially produced ROS [133,134]. Fructose intake diminishes the antioxidant machinery of mitochondria, increasing oxidative stress, which causes the lipid peroxidation of polyunsaturated fatty acids, and allows the attack of free radicals on mitochondrial DNA; as a result, mitochondrial biogenesis is also affected [133]. Mitochondrial dysfunction results in low fatty acid oxidation, decreased hepatic ATP levels, and increased hepatic oxidative stress [135,136]. All these effects are, at least in part, regulated through PPAR-α inhibition. On the other hand, fructose oxidation also produces carbonyl compounds such as glycolaldehyde, a metabolite of glyceraldehyde, and glyoxal, the major product of glycolaldehyde oxidation, which is associated with cellular injury and dysfunction, including the inhibition of mitochondrial respiration and induction of mitochondrial permeability transition, leading to cell death [33,67,137].

Additionally, the consumption of fructose but not glucose increases apolipoprotein CIII through the ChREBP pathway, increasing triglyceride and low-density lipoprotein levels upon fructose metabolism, and represents a significant contributor to cardiometabolic risk [138,139]. These observations suggest that ChREBP plays an important role in the pathogenesis of NASH; however, the suggested protective role of ChREBP deserves further investigation [127].

#### 2.3.5. Sterol-Responsive Element-Binding Protein and Fructose

The SREBP protein is generated in the endoplasmic reticulum as a complex with SREBP cleavage-activating protein (SCAP). SREBP1c is mainly produced in the liver and is activated by changes in nutritional status [140]. As in the intestine, fructose in the liver also contributes to increasing SREBP1c expression, which plays a pivotal role in lipid metabolism [138,141]. The deleterious effects on lipid metabolism of excessive fructose consumption are fasting and postprandial hypertriglyceridemia, and increased hepatic synthesis of lipids, very-low-density lipoproteins (VLDLs), and cholesterol [138,139,142,143]. It has been shown that the elevated levels of plasma triacylglycerol during high fructose feeding may be due to the overproduction and impaired clearance of VLDL, and chronic oxidative stress potentiates the effects of high fructose on the export of newly synthesized VLDL [144]. Moreover, in humans diets high in fructose have been observed to reduce postprandial serum insulin concentration; therefore, there is less stimulation of lipoprotein lipase, which causes a greater accumulation of chylomicrons and VLDL because lipoprotein lipase is an enzyme that hydrolyzes triglycerides in plasma lipoproteins [145]. High fructose consumption induces the hepatic transcription of hepatocyte nuclear factor 1, which upregulates aldolase B and cholesterol esterification 2, triggering the assembly and secretion of VLDL, resulting in the overproduction of free fatty acids [146]. These free fatty acids increase acetyl-CoA formation and maintain NADPH levels and NOX activation [146]. NOX, which uses NADPH to oxidize molecular oxygen to the superoxide anion [140], and xanthine oxidoreductase (XO), which catalyzes the oxidative hydroxylation of hypoxanthine to xanthine and xanthine to uric acid, are the main intracellular sources of ROS in the liver [147,148]. NOX reduces the bioavailability of nitric oxide and thus impairs the hepatic microcirculation and promotes the proliferation of HSCs, accelerating the development of liver fibrosis [147,148]. ROS derived from NOX lead to the accumulation of unfolded proteins in the endoplasmic reticulum lumen, which increases oxidative stress [146].

In hepatocytes, cytoplasmic Ca^2+^ is an important regulator of lipid metabolism. An increased Ca^2+^ concentration stimulates exacerbated lipid synthesis [145]. A high fructose intake induces lipid accumulation, leading to protein kinase C phosphorylation, stressing the endoplasmic reticulum [149]. Elevated activity of the protein kinase C pathway has been reported to stimulate ROS-generating enzymes such as lipoxygenases. A prolonged endoplasmic reticulum stress response activates SREBP1c and leads to insulin resistance [140,150]. Calcium signaling is also important for liver regeneration, and increased intracellular calcium homeostasis is known to be involved in tumor initiation, progression, and metastasis; therefore, the alteration of calcium homeostasis by high fructose consumption could be an important mechanism in the development of cancer [151,152].

Some evidence indicates that there is a synergy between SREBP activation with the stimulation of the inflammatory pathway mediated by NF-κB and cholesterol homeostasis. Activated NF-κB increases SCAP expression, resulting in the activation of the SCAP–SREBP complex, triggering an exacerbated inflammatory response and cholesterol accumulation [153]. Furthermore, some reports indicate that fructose supplementation leads to insulin receptor downregulation because protein-tyrosine phosphatase 1B activity decreases the phosphorylation of the insulin receptor and induces protein phosphatase 2A, increasing SREBP1c, aggravating hepatic insulin resistance via intricate metabolic pathways [88]. Extensive reviews have been published on the lipogenic effect of fructose [70,135,154]; however, the deleterious effects of fructose in the liver go beyond the steatotic effect. Hepatic cholesterol accumulation is associated with inflammatory cell infiltration [155]. Dietary fructose induces strong SREBP1c activation, and the consequent palmitate production causes lipotoxicity in the endoplasmic reticulum; these events are the leading factors responsible for the greater Nrf2 inhibition and more intense hepatic inflammatory response driven by NLRP3 inflammasome activation [107,156].

Some authors have proposed “multiple parallel hit” theories to explain the development of the metabolic disease NAFLD, the first hit being the accumulation of fat in the liver (mainly triglycerides), followed by multifactorial processes that involve oxidative stress, inflammation, and hyperuricemia as the main factors [157,158]. DNA methylation is an epigenetic mechanism that decreases gene expression. Accumulating evidence suggests that excessive fructose intake drives epigenetic alterations, including the hypermethylation of the carnitine palmitoyl transferase 1A and PPAR-α genes [159,160]. Increased malonyl-CoA, which is synthesized by the enzyme acetyl-CoA carboxylase, inhibits carnitine palmitoyl transferase 1A, which is the rate-limiting step of the oxidation of lipids in the mitochondria, leading to the disruption of β-oxidation and accumulation of hepatic lipids, particularly fatty acids such as diacylglycerols and ceramides, which inhibit the insulin signaling pathway through protein kinase C activation and the inhibition of the protein kinase AKT, respectively [102,160] (Figure 3).

This scenario can be worsened because high-glycemic diets induce the conversion of glucose to fructose by the aldose reductase enzyme. Fructose can be endogenously synthesized in the body via the polyol pathway, a two-step conversion of glucose to fructose, which is relatively inactive under physiological conditions [161,162]. In addition, in high-glycemic diets, glucose is metabolized by fructose-3-phosphokinase to a highly reactive molecule, fructose-3-phosphate, causing the formation of advanced glycation end products, which can trigger inflammatory pathways through the activation of signaling pathways such as NF-κB and mitogen-activated protein kinases, aside from increasing lipogenesis and the disruption of β-oxidation, independently of caloric intake and weight gain [135,163,164]. On the other hand, fructose can be released from the liver to the systemic circulation and filtered and excreted by the kidneys, a decisive organ for fructose disposal, increasing metabolic abnormalities [165,166].

#### 2.3.6. Uric Acid and Fructose

KHK utilizes ATP to phosphorylate fructose to form fructose-1-phosphate, leading to intracellular phosphate exhaustion, which in turn activates the enzyme adenosine monophosphate (AMP) deaminase, which converts AMP to inosine monophosphate (IMP) (Figure 3). Consequently, xanthine is formed by the XO enzyme, which is associated with the production of large amounts of highly cytotoxic ROS as well as hydrogen peroxide [72]. Chronic fructose consumption stimulates purine nucleotide turnover, which culminates in the synthesis of uric acid from xanthine by XO, leading to uric acid accumulation within hepatic cells [167]. 4-Hydroxynonenal, which is formed by the attack of ROS on biological membranes, induces XO, a key enzyme in purine and free radical metabolism; in turn, high activity of XO may further promote oxidative stress in the liver [168,169]. Increased systemic oxidative stress is recognized as an essential cause of elevated uric acid and inflammation [170]. Uric acid is a potent inducer of the inflammatory response by activating NF-κB through inducing the phosphorylation of IKK and IκBα, followed by the subsequent stimulation of NF-κB activity [108,171,172]. The systemic effects of uric acid are related to endothelial injury and dysfunction. Uric acid directly inhibits endothelial nitric oxide synthase; the impairment of nitric oxide synthesis decreases vascular smooth muscle relaxation and increases systolic blood pressure, leading to hypertension [173].

Uric acid also promotes fat synthesis within hepatocytes through the translocation of the NADPH oxidase subunit 4 to the mitochondria, increasing superoxide formation [174]. In turn, this increase in ROS inhibits the enzyme aconitase, which catalyzes the conversion of citrate to isocitrate in the mitochondrial matrix in the Krebs cycle and promotes citrate accumulation; then, citrate is converted to acetyl-CoA for de novo lipogenesis by FASN [174]. Oxidative stress and uric acid are amplifying actors that activate the nuclear factor of activated T cells, which plays a role in the regulation of inflammation and upregulates aldose reductase via the polyol pathway, leading to hepatic steatosis [162]. Oral and coworkers (2019) found a positive correlation between the degree of liver damage and uric acid concentration in non-obese and young patients with NAFLD, who had higher uric acid concentrations than the healthy control group [175]. Some reports showed that uric acid in the liver promotes oxidative stress and inflammation through the inhibition of Nrf2 and overproduction of thioredoxin, which results in NLRP3 inflammasome activation [103,176,177]. In NAFLD patients who have fatty liver but no inflammation, the expression of NLRP3 components is increased, but the inflammasome is not activated; however, in patients with fatty liver and inflammation, the NLRP3 inflammasome complex is assembled and functional [106]. Therefore, the activation of the NLRP3 inflammasome is associated with liver disease progression from simple fatty liver to NASH with inflammation and fibrosis [106]. Additionally, increased serum uric acid levels are indicative of a greater probability of elevated serum alanine aminotransferase and gamma-glutamyl transferase, two markers of hepatic necroinflammation, and are associated with a greater risk of cirrhosis-related hospitalization or death [64]. These studies support the role of uric acid as a risk marker of liver damage via NLRP3 inflammasome activation; moreover, it represents a non-invasive marker and a possible predictor of NASH. These findings suggest that activation of NLRP3 inflammasome induces a fibrogenic micro-environment in the liver. Therefore, the inhibition of NLRP3 inflammasome is a promising therapeutic tool to ameliorate hepatic fibrosis. In addition, some antioxidants have been shown to block the NRLP3 inflammasome signaling pathway and thus may be helpful to decrease NASH development.

#### 2.3.7. MicroRNAs and Fructose

Novel evidence suggests that miRNAs play an important role in liver health and disease. The expression of miRNAs can be modified by increasing fructose intake and/or uric acid production. Rats fed a high-fructose diet have decreased miRNA-122, miRNA-451, and miRNA-27a compared to control-fed rats [178]. Additionally, miRNAs in mice such as miRNA-34a, miRNA-335, miRNA-221, and miRNA-9 are upregulated in the liver by high fructose intake [179]. There is cumulative evidence that some miRNAs regulate several signaling pathways, leading to oxidative stress and inflammation in the liver. For example, in humans the elevation of miR-214 levels decreases glutathione reductase and cytochrome P450 activities; consequently, hepatic oxidative stress is augmented [180]. The attenuation of miRNA-199a-5p produces apoptosis associated with endoplasmic reticulum stress [181]. miRNA-223 is expressed in the liver and prevents inflammation, the activation of HSCs, and fibrosis through disrupting the activation of the NLRP3 inflammasome [182]. In addition, it has been observed through an in vitro transfection assay that miRNA-33 is responsible for the regulation of SREBP1 after fructose ingestion [183]. Mice with miRNA-29a overexpression show decreased DNA oxidative damage in an NAFLD model, suggesting its role in neutralizing oxidative stress [184]. Furthermore, miRNA-29a contributes to a reduction in NF-κB activity, which leads to a decrease in the inflammatory process and provides protection against fibrosis by suppressing TGF-β and SMAD3, the canonical signaling pathway for HSC activation. MiRNA-149-5p is induced by uric acid in hepatocytes, causing lipid accumulation via the upregulation of FGF21, a protein implicated in lipid metabolism that is considered an anti-metabolic-syndrome hormone, therefore playing an important role in the prevention of NAFLD development [57].

#### 2.3.8. Cancer and Fructose

In 1924, Otto Warburg described that cancer cells could obtain energy by fermenting glucose into lactate, and this is called the “Warburg effect” [185]. Fructose promotes the Warburg effect, increasing glycolysis and suppressing fat oxidation, which may promote mitochondrial dysfunction, tumor growth, and metastasis [185]. Fructose-rich diets can increase HCC incidence because it was found that fructokinase and Glut5 are highly expressed in diverse types of cancer, and that the upregulation of Glut5 correlates with a poor prognosis in HCC [186,187]. Importantly, several investigators have suggested that high fructose intake not only promotes cancer development in various tissues but also proposed that endogenously produced fructose in cancer cells could potentially stimulate cancer growth [185]. The key enzyme that stimulates endogenous fructose production is aldose reductase in the polyol pathway. Fructose also induces metabolic changes via KHK-A, promoting the pentose phosphate pathway, the development of HCC [188], and the serine-to-glycine synthesis pathway for HCC growth [189]. Notably, fructose can be utilized by cancer cells as an energy source and, subsequently, for the synthesis of nucleic acids through the pentose phosphate pathway. Fructose also promotes colon cancer metastasis to the liver via the KHK–aldolase B pathway, and a high-fructose diet increases colorectal liver metastasis [190]. The silencing of aldolase B or the restriction of fructose in the diet suppresses liver metastasis from colorectal cancer [190,191]. Moreover, as mentioned above, uric acid is a by-product of fructose metabolism that stimulates the production of mitochondrial ROS and aldolase. In clinical studies, high uric acid is considered a significant risk factor for active hepatocarcinogenesis [191]. Fructose metabolism during carcinogenesis elevates oxidative stress and inflammation [192]. However, the effects of endogenous or exogenous fructose in cancer need to be investigated in more detail.

## 3. Conclusions and Perspectives

Research on the impact of human nutrition on health and disease is vast. However, the molecular mechanisms involved in nutrition’s effects on human diseases are far from being fully understood. Plenty of evidence indicates that fructose and its metabolites play a significant role in the development of liver disease. The multiple mechanisms that fructose triggers have placed it in the eye of the hurricane in metabolic disorders of the liver. Although direct extrapolation from animal findings to humans is not recommended, basic research has illuminated some of the cellular and molecular mechanisms that are involved in the deleterious effects of the overconsumption of fructose, including oxidative stress, inflammation, higher serum uric acid levels, hypertriglyceridemia, higher systolic blood pressure, insulin resistance, fibrosis, cirrhosis, and HCC. Fructose-induced hepatic injury depends strongly on the activation of lipogenesis and inflammatory signaling pathways, which, in turn, trigger fibrosis and HCC development. Free radical and uric acid overproduction induced by excessive fructose consumption also play pivotal roles in fatty liver, inflammation, fibrosis, and HCC progression through a variety of signaling pathways. These observations provide mechanistic information on NASH development and may be used for the development of new drugs and therapies. Several anti-inflammatory, antifibrotic, and anticancer targets are now known in the pathogenic pathways involved in fructose overconsumption. However, more in-depth studies dealing with the involved molecular mechanisms of fructose-driven fibrogenesis are required to find new therapeutic targets for drug development to prevent hepatic fibrosis.

The alarming increase in metabolic syndrome and comorbidities can only be attenuated if the consumption of fructose, mainly in soft beverages, is significantly reduced worldwide. In addition, an active lifestyle incorporating the practice of sports seems to be useful for fighting the sedentarism associated with obesity. Patients suffering from hepatic maladies should be recommended to reduce fructose consumption to prevent aggravation of their condition because fructose may act as a conjoint pathological agent.

## Figures and Tables

**Figure 1 ijms-22-06969-f001:**
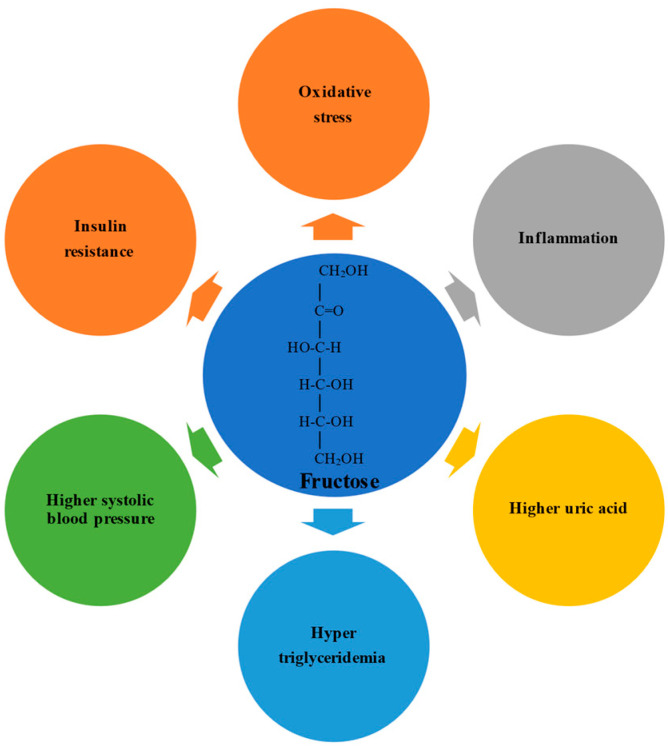
Systemic effects of overconsumption of fructose. Elevated fructose intake is implicated in increased oxidative stress, inflammation, higher uric acid levels, hypertriglyceridemia, higher systolic blood pressure, and insulin resistance, which are associated with the development or worsening of liver diseases.

**Figure 2 ijms-22-06969-f002:**
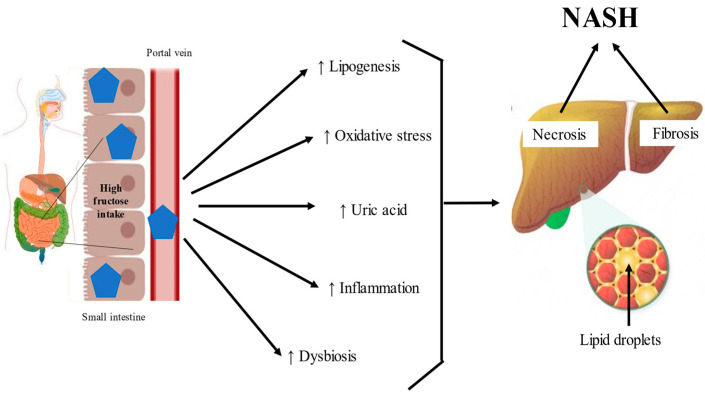
Fructose’s effects on the gut. Excessive fructose intake induces lipogenesis, oxidative stress, uric acid production, inflammation, and dysbiosis on the gut, which trigger necrosis and fibrosis in nonalcoholic steatohepatitis (NASH).

**Figure 3 ijms-22-06969-f003:**
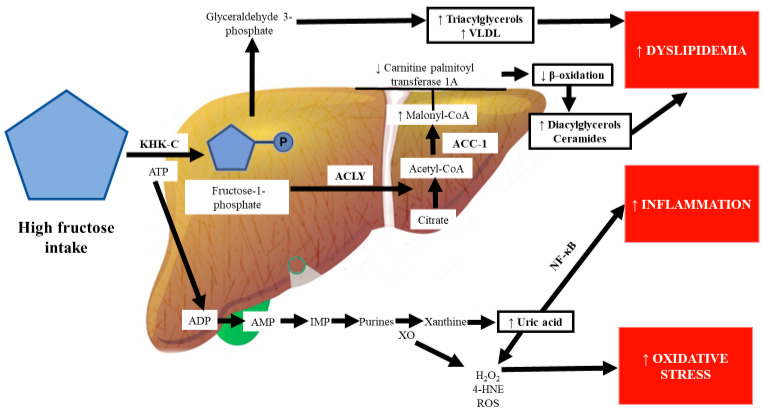
Metabolism of fructose in the liver. Fructose enters hepatocytes and is phosphorylated to fructose 1-phosphate by ketohexokinase C (KHK-C). Fructose 1-phosphate is converted to glyceraldehyde-3 phosphate and is a substrate for the synthesis of triacylglycerols and very-low-density lipoproteins (VLDL). Fructose intake can upregulate the ATP citrate lyase (ACLY) enzyme, promoting citrate breakdown to acetyl-CoA. Acetyl-CoA is converted to malonyl-coenzyme A (malonyl-CoA) by acetyl-CoA carboxylase (ACC-1). Elevated levels of malonyl-CoA inhibit β-oxidation through limiting carnitine palmitoyl transferase 1A, which promotes the accumulation of diacylglycerols and ceramides, causing hepatic dyslipidemia. Moreover, increased KHK-C activity exhausts adenosine triphosphate (ATP), generating adenosine diphosphate (ADP), which is converted to adenosine monophosphate (AMP). In turn, AMP is transformed to inosine monophosphate (IMP), increasing purine production. Xanthine oxidoreductase (XO) produces oxygen reactive species (ROS), hydrogen peroxide (H2O2), 4-hydroxynonenal (4-HNE), and xanthine. Then, xanthine is metabolized, resulting in the overproduction of uric acid and ROS, which induce oxidative stress. Uric acid activates nuclear factor-κB (NF-κB), triggering inflammation.

**Figure 4 ijms-22-06969-f004:**
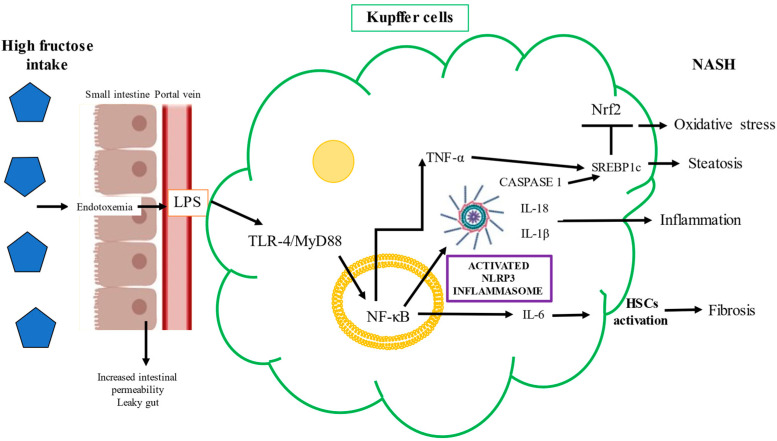
Molecular mechanisms by which fructose induces nonalcoholic steatohepatitis. Increased intestinal permeability (“leaky gut”) and dysbiosis produced by high fructose intake promote lipopolysaccharide (LPS) translocation from the intestine to the portal blood to reach the liver. Then, LPS activates the Toll-like receptor (TLR)-4/MyD88 signaling pathway, inducing tumor necrosis factor-alpha (TNF-α) through the nuclear translocation of transcription nuclear factor kappa B (NF-κB), which reinforces the inflammatory process through NLRP3 inflammasome activation and the subsequent maturation of interleukin (IL)-1 beta (β), caspase 1, and IL-18. Additionally, TNF-α and caspase 1 promote sterol-responsive element-binding protein 1 c (SREBP1c) activation and nuclear factor E2-related factor 2 (Nrf2) inhibition, while IL-6 drives hepatic stellate cell (HSC) activation, an orchestrated interaction of various molecular factors, leading to oxidative stress, inflammation, steatosis, and fibrogenesis, which pave the way to nonalcoholic steatohepatitis (NASH) development.

**Figure 5 ijms-22-06969-f005:**
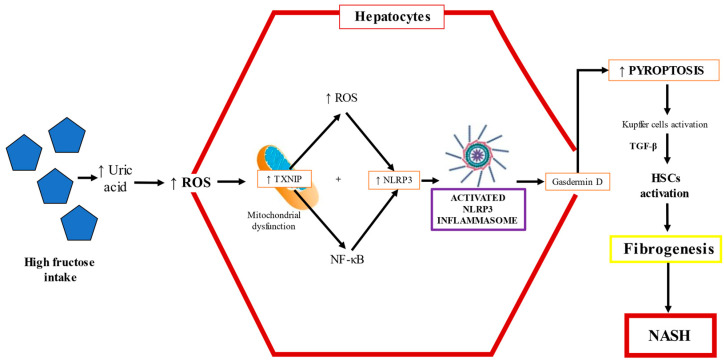
The fructose–pyroptosis axis. High fructose intake induces uric acid production in the intestine and liver, increasing reactive oxygen species (ROS). The resulting oxidative stress promotes the intracellular translocation of the thioredoxin-interacting protein (TXNIP) in the mitochondria; then, the interaction between TXNIP and NOD-like receptor family pyrin domain containing 3 (NLRP3) leads to NLRP3 inflammasome activation. The assembly of the inflammasome machinery enhances the pyroptosis of hepatocytes by the gasdermin D pathway and leads to the activation of Kupffer cells and transforming growth factor-beta (TGF-β) secretion, which results in HSC activation, triggering fibrogenesis in nonalcoholic steatohepatitis (NASH).

## Data Availability

Not applicable.

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
