# Peer review of "Fructose and the Liver"

_ijms, 2021, doi:10.3390/ijms22136969_

Round 1
Reviewer 1 Report
This is a timely and relevant review on the effect of fructose on the liver. The increasing consume of fructose, free or as a part of fructose, is an increasingly health problem, mainly because of the inexpedient effects of fructose in the liver. In this review the authors describe the essential findings of fructose in the body with emphasis on the liver. The authors cite the important comprehensive literature in the field and the illustrations are relevant.
However, the description of the literature and our current knowledge has character of a continuous listing of theories without a real discussion. It would be appropriate with a deeper discussion of data and the author should somehow try to take position on what the dominating theories are. Without that the review is difficult to read and to use as an introduction into he field.
In line with with that the review does not really discuss the problems with the differences between species handling fructose. Many theories on biochemical mechanisms in the liver is based on experimental animal studies, and it obviously not certain that these findings can be translated to humans. Neither is the problems in humans rarely only a fructose problem because of comorbidity and high uptake of other components with toxic effect on the liver such as ethanol and high caloric intake in general. This needs to more extensively discussed in relation to development of NAFLD and NASH.
Minor point.
The effect of non-absorbed fructose in the intestine is described as a special toxic effects of fructose. In fact, this is just the classical systems of osmotic diarrhea as also seen in lactose intolerance and when glucose uptake is inhibited.
Author Response
REPLY TO REVIEWER 1
Reviewer 1
Reviewer 1. This is a timely and relevant review on the effect of fructose on the liver. The increasing consume of fructose, free or as a part of fructose, is an increasingly health problem, mainly because of the inexpedient effects of fructose in the liver. In this review the authors describe the essential findings of fructose in the body with emphasis on the liver. The authors cite the important comprehensive literature in the field and the illustrations are relevant.
Answer: Thank you for your comments.
Reviewer 1. However, the description of the literature and our current knowledge has character of a continuous listing of theories without a real discussion. It would be appropriate with a deeper discussion of data and the author should somehow try to take position on what the dominating theories are. Without that the review is difficult to read and to use as an introduction into he field.
Answer: Answer: Thank you very much for this recommendation. The manuscript was improved according to your comments. Discussion of data was increased and a position was taken.
Please see lines 230-235 and 583-587 on pages 6 and 16, respectively, highlighted text.
Reviewer 1. In line with with that the review does not really discuss the problems with the differences between species handling fructose. Many theories on biochemical mechanisms in the liver is based on experimental animal studies, and it obviously not certain that these findings can be translated to humans.
Answer: We appreciate your comments. Effectively, direct extrapolation from animals to humans is not recommended. Therefore, the manuscript was revised, and corrections were made accordingly.
For example, please see the lines 642-643 and 655-657, on page 18. Corrections are highlighted.
Reviewer 1. Neither is the problems in humans rarely only a fructose problem because of comorbidity and high uptake of other components with toxic effect on the liver such as ethanol and high caloric intake in general. This needs to more extensively discussed in relation to development of NAFLD and NASH.
Answer: We agree with the comment of the reviewer. Human liver diseases have diverse etiologies i.e., ethanol drinking, viruses, biliary track obstruction, etc. We think that patients suffering from hepatic maladies should be recommended to reduce fructose consumption to prevent aggravation of their condition because fructose may act as a conjoint pathological agent.
Please see the lines 336-338 and 661-663 on pages 10 and 18, respectively. Corrections are highlighted.
Reviewer 1
Minor point.
The effect of non-absorbed fructose in the intestine is described as a special toxic effects of fructose. In fact, this is just the classical systems of osmotic diarrhea as also seen in lactose intolerance and when glucose uptake is inhibited.
Answer: We appreciate the comments of the reviewer. The paragraph was removed.
Reviewer 2 Report
The review was generally well-written, but the introduction needs to be re-organized and introduce what will be reviewed and discussed by the authors. Additionally, when the authors cite experimental evidence, it would be helpful in indicate what model system was used in the study (type of organism, strain, in vivo or in vitro), since we know that certain metabolic reactions and pathways are animal model-dependent.
1. Introduction
The introduction should be better organized and introduce the review content overall.
Authors discuss the prevalence of MetS globally. It might be worth mentioning the range of prevalence in different regions since it is known to be different.
Line 47-48. Please also cite the original paper where this was found.
Line 46: This paragraph starts to mention fructose is a mediator of NAFLD but then returns to talk about HCC in the end. The next paragraph returns to talking about fructose. There needs to be a better organization and transition between the paragraphs. Consider moving sentences in Line 46-48 to the end of the paragraph or beginning of the next paragraph.
2. Deleterious metabolic effects of fructose
Overall, this is a very large main section even though there are subsections. If possible, I would suggest further dividing this into main sections.
Section 2.2
This section has very large paragraphs and should be divided.
Line 124: Does this refer to just mice or is this also found in other animals? The citation used is using a mouse model.
Line 148: “Uric acid, a weak organic acid end-product of purine metabolism”. Since this is species specific, please indicate which species this is referring to.
Line 170-173: This sentence seems out of place. Consider removing or moving to another part of the section.
Figure 2 is referred to in line 220 a sentence talking about gut dysbiosis in section 2.2 titled “Fructose in the intestine” however, the figure is about metabolism of fructose in the liver which are mentioned specifically in later sections. I would change this figure to better display the overall rationale for this section. For this reviewer, this figure might make more sense in later sections.
Increase font size in Figure 2. The text is quite small.
Section 2.3
Figure 3 is referred to in a sentence line 248-250 which is talking about a specific metabolic reaction, but this is not seen in the figure. I would suggest aligning the figures better to the text.
Line 312: Indicate which models specifically.
Line 567-570: The first sentence is referring to what is observed in rats and the next is citing a study in mice. The authors should state the model the observation was seen in.
Section 2.3.5. is quite long and seems to go into other important pathways. For example, one paragraph focuses on calcium signaling and authors only moderately connect it to SREBP. The authors also later talk about Ahr, epigenetic changes and kidney. The authors should better focus this section or better transition the different paragraphs to link them to the central theme of the section.
Line 462-464: Verify the citation.
Author Response
REPLY TO REVIEWER 2
Reviewer 2
- Introduction
The introduction should be better organized and introduce the review content overall.
Answer: Thank you for your suggestion. Please see the lines: 53-54 on page 2. Text is highlighted.
In general, manuscript was better organized.
Authors discuss the prevalence of MetS globally. It might be worth mentioning the range of prevalence in different regions since it is known to be different.
Answer: Thank you for your suggestion. It was done. Please see the lines: 31-32 on page 1. Text is highlighted.
Reviewer 2: Line 47-48. Please also cite the original paper where this was found.
Answer: Thank you for your suggestion. It was done. Please see lines 56-57 on page 2. Original article 24 was added to the reference list according to the recommendation. Correction is highlighted.
Reviewer 2: Line 46: This paragraph starts to mention fructose is a mediator of NAFLD but then returns to talk about HCC in the end. The next paragraph returns to talking about fructose. There needs to be a better organization and transition between the paragraphs. Consider moving sentences in Line 46-48 to the end of the paragraph or beginning of the next paragraph.
Answer: Thank you for your suggestion. It was done. Please see lines 55-56 on page 2. In general, manuscript was better organized.
Corrections are highlighted.
Reviewer 2: 2. Deleterious metabolic effects of fructose
Overall, this is a very large main section even though there are subsections. If possible, I would suggest further dividing this into main sections.
Answer: We think that this section is not too long. We better divide Section 2.2. Please see below.
Reviewer 2: Section 2.2
This section has very large paragraphs and should be divided.
We appreciate the comments of the reviewer. Corrections were performed and highlighted. Please see Sections: 2.2.1., 2.2.2., 2.2.3., 2.2.4.
Line 124: Does this refer to just mice or is this also found in other animals? The citation used is using a mouse model.
Answer: We appreciate the comments of the reviewer. Please see line 124-135 on page 4. Correction is highlighted.
Reviewer 2: Line 148: "Uric acid, a weak organic acid end-product of purine metabolism". Since this is species specific, please indicate which species this is referring to.
Answer: Thank you for your comment. The specie is humans. Please see line 156 on page 4. Correction is highlighted.
Reviewer 2: Line 170-173: This sentence seems out of place. Consider removing or moving to another part of the section.
Answer: Thank you for your comment. The sentence was removed.
Reviewer 2: Figure 2 is referred to in line 220 a sentence talking about gut dysbiosis in section 2.2 titled "Fructose in the intestine" however, the figure is about metabolism of fructose in the liver which are mentioned specifically in later sections. I would change this figure to better display the overall rationale for this section. For this reviewer, this figure might make more sense in later sections.
Answer: Thank you for your comment. Figures were corrected. Please see new figures 2 and 3.
Reviewer 2: Increase font size in Figure 2. The text is quite small.
Answer: Thank you for your comment. The font size was increased. Please see new Figure 2.
Reviewer 2: Section 2.3
Figure 3 is referred to in a sentence line 248-250 which is talking about a specific metabolic reaction, but this is not seen in the figure. I would suggest aligning the figures better to the text.
Answer: We appreciate the comments of the reviewer. We are very sorry for this mistake; Figure 3 was misplaced. This was corrected
Reviewer 2: Line 312: Indicate which models specifically.
Answer: Thank you for your comment. It was done. Please see line 311 on page 9. Correction is highlighted.
Reviewer 2: Line 567-570: The first sentence is referring to what is observed in rats and the next is citing a study in mice. The authors should state the model the observation was seen in.
Answer: We appreciate the comment of the reviewer. The animal model is mice. Please see line 593 on page 17. Correction is highlighted.
Reviewer 2: Section 2.3.5. is quite long and seems to go into other important pathways. For example, one paragraph focuses on calcium signaling and authors only moderately connect it to SREBP. The authors also later talk about Ahr, epigenetic changes and kidney. The authors should better focus this section or better transition the different paragraphs to link them to the central theme of the section.
Answer: We appreciate the reviewer's comments and agree that this section is too long; therefore, it was shortened and reordered. Please see the new Section 2.3.5, where the original text (original lines 518-520) was eliminated. Please see lines 536-538 on page 15. Corrections are highlighted.
In addition, the text about Ahr (original lines 599-505) was moved to section 2.3.3 (lines 405-411 on page 12 of the new version of the manuscript). We think that the new corrected version of the manuscript is better connected. Corrections are highlighted.
Reviewer 2: Line 462-464: Verify the citation.
Answer: Thank you for your comment. The citation was corrected. Please see line 492 on page 14. Correction is highlighted.